# Kinsenoside Suppresses DGAT1-Mediated Lipid Droplet Formation to Trigger Ferroptosis in Triple-Negative Breast Cancer

**DOI:** 10.3390/ijms26052322

**Published:** 2025-03-05

**Authors:** Yaqin Yang, Dandan Chen, Yuru Zhu, Min Zhang, Huajun Zhao

**Affiliations:** 1School of Pharmaceutical Sciences, Zhejiang Chinese Medical University, Hangzhou 311402, China; rc.yaqin0902@zcmu.edu.cn (Y.Y.); 202321113911035@zcmu.edu.cn (D.C.); zyr@zcmu.edu.cn (Y.Z.); zmandwww@163.com (M.Z.); 2Academy of Chinese Medical Sciences, Zhejiang Chinese Medical University, Hangzhou 310053, China

**Keywords:** kinsenoside, triple-negative breast cancer, ferroptosis, lipid droplet formation, DGAT1

## Abstract

Triple-negative breast cancer (TNBC) presents limited therapeutic options and is characterized by a poor prognosis. Although Kinsenoside (KIN) possesses a wide range of pharmacological activities, its effect and mechanism in TNBC remain unclear. The objective of this research was to explore the therapeutic effectiveness and the molecular mechanisms of KIN on TNBC. Xenograft experiment was carried out to assess the impact of KIN on TNBC in vivo. The effect of KIN on TNBC in vitro was evaluated through the analysis of cell cytotoxicity and colony formation assays. Oil Red O staining and BODIPY 493/503 fluorescence staining were employed to detect the effect of KIN on lipid droplet (LD) formation. Transcriptomics and inhibitor-rescue experiments were conducted to investigate the role of KIN on TNBC. Mechanistic experiments, including quantitative real-time polymerase chain reaction (RT-qPCR), Western blotting, diacylglycerol acyltransferase 1 (DGAT1) overexpression assay, and flow cytometric assay, were employed to uncover the regulatory mechanisms of KIN on TNBC. KIN inhibited tumor growth without causing obvious toxicity to the liver and kidneys. In vitro experiments demonstrated that KIN significantly inhibited the viability and proliferation of TNBC cells, accompanied by decreased LD formation and lipid content. Polyunsaturated fatty acids (PUFAs) levels were significantly increased by KIN. Furthermore, transcriptomics and inhibitor-rescue experiments revealed that KIN induced ferroptosis in TNBC cells. KIN could significantly regulate ferroptosis-related proteins. Lipid peroxidation, iron accumulation, and GSH depletion also confirmed this. The LD inducer mitigated the KIN-induced ferroptosis in TNBC. The overexpression of DGAT1 attenuated the effects of KIN on cell viability and proliferation. Furthermore, the overexpression of DGAT1 inhibited the effect of KIN to trigger ferroptosis in TNBC cells. Our findings confirmed that KIN could trigger ferroptosis by suppressing DGAT1-mediated LD formation, thereby demonstrating a promising therapeutic effect of KIN in TNBC.

## 1. Introduction

Triple-negative breast cancer (TNBC) is a breast cancer subtype without human epidermal growth factor receptor 2 (HER-2), estrogen receptor (ER), and progesterone receptor (PR) expression [1]. Due to these characteristics, treatment options for TNBC are limited. Therefore, there is an urgent need to develop new therapeutic strategies targeting TNBC.

Ferroptosis is a form of cell death induced by iron-dependent lipid peroxidation promoted by the presence of polyunsaturated fatty acids (PUFAs) [2]. TNBC is more sensitive to ferroptosis than other types of breast cancer [3,4]. Therefore, targeting ferroptosis in TNBC cells is a promising therapeutic approach.

An increasing number of studies indicate that the occurrence and development of cancer are often accompanied by disorders of lipid metabolism [5,6]. Lipid droplet (LD) functions as a cellular organelle dedicated to lipid storage [7]. The biological process of lipid droplet formation requires the coordinated action of various proteins, among which diacylglycerol acyltransferase 1 (DGAT1) is a key protein that promotes LD formation [8]. It was reported that the number of LD within tumor cells are often present at a high level, especially in TNBC [9]. Lorito N et al. have confirmed that the excessive accumulation of LD in TNBC enhances tumor invasiveness [10]. The formation of LD is closely related to the dynamic balance of PUFAs concentrations. Inhibiting LD formation can block the storage of PUFAs, ultimately leading to cellular peroxidation and ferroptosis [11]. Therefore, inducing ferroptosis in TNBC cells by pharmacologically inhibiting LD formation provides a new perspective for the treatment of TNBC.

In recent years, traditional Chinese medicine (TCM) and natural products have been widely used in the treatment of cancer, characterized by their high efficacy and low toxicity [12]. *Anoectochilus roxburghii*, also known as Jin Xian Lian, refers to the fresh or dried whole plant of *Anoectochilus roxburghii* (Wall.) Lindl [13]. *Anoectochilus roxburghii* has a broad anti-tumor effect, including breast cancer [14] and colon cancer [15]. Its lipid-lowering effect was also reported [16]. In addition, the polysaccharides from *Anoectochilus roxburghii* were confirmed to effectively inhibit the proliferation of breast cancer MCF-7 cells [17]. Kinsenoside (KIN), a major active compound in *Anoectochilus roxburghii,* has pharmacological effects such as lipid-lowering [18,19], antioxidant [20], and liver-protecting properties [21]. However, the therapeutic effect and mechanism of KIN on TNBC are still unclear.

This study aims to evaluate the therapeutic potential of KIN on TNBC in vitro and in vivo. In addition, we explored the molecular mechanism by which KIN inhibits lipid droplet formation and induces ferroptosis in TNBC cells. This research provides strong evidence for the treatment of TNBC with KIN and offers important scientific support for TCM in treating TNBC by targeting LD formation to induce ferroptosis.

## 2. Results

### 2.1. KIN Inhibited Tumor Growth in Mouse Xenograft Tumor Models

To confirm the therapeutic efficacy of KIN against TNBC, the drug efficacy evaluation experiment of a xenograft mouse model was performed. After adaptive feeding, an orthotopic MDA-MB-231 breast tumor xenograft model was established, and the animal experimental protocol is shown in Figure 1A. Compared with the control group, KIN significantly decelerated tumor growth in a dose-dependent manner (Figure 1B–D). The immunohistochemistry (IHC) experiment results indicated that KIN inhibited the expression of Ki67 in tumor tissues significantly (Figure 1E). The body weights and organ pathology of mice treated with KIN were not affected significantly, indicating that there is no apparent toxicity after KIN’s treatment (Figure 1F,G). The measurement of serum alanine aminotransferase (ALT), aspartate aminotransferase (AST), blood urea nitrogen (BUN), and Creatinine (CRE) levels also confirmed this result (Figure 1H–K). These findings suggest that KIN significantly suppresses tumor growth without obvious toxicity.

### 2.2. KIN Inhibits Growth of TNBC Cells In Vitro

We first evaluated the effects of various concentrations of KIN (0, 10, 20, 40, 60, 80, and 100 μM) on the viability of TNBC (MDA-MB-231 and MDA-MB-468) and normal breast epithelial (MCF-10A) cells using an MTT assay. As shown in Figure 2A–C, KIN inhibited the viability of TNBC cells in a dose-dependent manner without affecting normal mammary epithelial cells MCF-10A. Furthermore, KIN administration significantly reduced colony formation compared with the control group (Figure 2D–F). In summary, the findings unequivocally showed that KIN markedly inhibited the growth of TNBC cells without exhibiting toxic effects on normal breast epithelial cells.

### 2.3. KIN-Induced Ferroptosis in TNBC Cells

To elucidate the mechanism of KIN-induced cell death, MDA-MB-231 cells treated with KIN were sent for transcriptomic analysis. Compared with the control group, a total of 128 differentially expressed genes were detected in the KIN-treated group (*p*-value < 0.01, fold change > 1.5). A volcano plot showed that 59 genes were downregulated and 69 genes were upregulated in the KIN-treated group (Figure 3A). Based on these differentially expressed genes, we conducted bioinformatics analysis to determine the key pathways altered by KIN. KEGG analysis revealed that the ferroptosis pathway was strongly activated after the application of KIN (Figure 3B). To ensure the reliability of the above transcriptomic results, inhibitor-rescue experiments were then conducted by administering the combined treatment with ferroptosis inhibitors in MDA-MB-231 and MDA-MB-468 cells. The results showed that ferroptosis inhibitor deferoxamine (DFO) could reverse the inhibitory effect of KIN on the viability of MDA-MB-231 and MDA-MB-468 cells to some extent (Figure 3C,D). A Western blotting experiment demonstrated that KIN could reduce the protein expression levels of GPX4 and SLC7A11 in vitro, while increasing the protein expression level of ACSL4 (Figure 3E,F). The same results were also observed in animal experiments (Figure 3G). Figure 3H–M demonstrated that KIN could significantly reduce glutathione (GSH) levels and elevate iron ion levels both in vitro and in vivo. Further evaluation showed that KIN significantly promoted lipid ROS elevation in TNBC cells (Figure 3N–Q). In summary, these results indicated that KIN could exert anti-TNBC properties by inducing ferroptosis.

### 2.4. KIN Inhibits Lipid Droplet Formation in TNBC Cells

An expanding dataset indicates a significant tumor-promoting function of increased lipid droplet buildup in a variety of cancers. As shown in Figure 4A–D, KIN significantly reduced the accumulation of triglycerides (TC) and total cholesterol (TG) in TNBC cells in a dose-dependent manner. In order to examine the modulatory function of KIN on lipid droplet formation in TNBC cells, we initially assessed LD formation using Oil Red O staining in MDA-MB-231 and MDA-MB-468 cells. Notably, KIN treatment decreased the accumulation of LD in the two TNBC cell lines (Figure 4E). Additionally, the cells were exposed to isopropanol to liberate oil droplets, revealing that KIN significantly reduced lipid buildup in TNBC cells compared to the control (Figure 4F,G). The accumulation of LD in the cell lines was also assessed using the fluorescent dye BODIPY 493/503. KIN treatment significantly reduced LD accumulation in MDA-MB-231 (Figure 4H,J) and MDA-MB-468 cells (Figure 4I,K) compared to the control group. When the formation of intracellular LD is inhibited, PUFAs will accumulate in large amounts within the cell. The presence of PUFAs can promote the occurrence of ferroptosis [22]. PUFAs quantification assay revealed an increased accumulation of PUFAs within the KIN treatment group (Figure 4L,M). These results confirmed that KIN could significantly inhibit the formation of LD in TNBC cells.

### 2.5. KIN Induces Ferroptosis in TNBC Cells by Inhibiting Lipid Droplet Formation

To elucidate the correlation between KIN’s inhibition of lipid droplet formation and induction of ferroptosis in TNBC cells, the lipid droplet inducer oleic acid (OA) was employed. The effect of different concentrations of OA on the viability of TNBC cells was assessed using the MTT method. The results showed that 100 μM OA had no significant effect on the viability of TNBC cells (Appendix A). As shown in Figure 5A,B, the increase in LD reduced the sensitivity of TNBC cells to KIN. This result was also confirmed in the colony formation assay (Figure 5C–E). Further research revealed that after treatment with the lipid droplet inducer OA, the ability of KIN to induce ferroptosis in TNBC cells was attenuated (Figure 5F,G). The inhibitory effect of KIN on the intracellular GSH levels in TNBC cells was also reversed by the lipid droplet inducer (Figure 5H,I). Similarly, the ability of KIN to increase intracellular iron levels in TNBC cells was also attenuated by the lipid droplet inducer (Figure 5J,K). These results confirmed that the mechanism by which KIN induced ferroptosis in TNBC cells was achieved by inhibiting lipid droplet formation.

### 2.6. KIN Inhibits the Expression of the Key Lipid Droplet Formation Protein DGAT1

To further uncover the mechanism by which KIN inhibits lipid droplet formation, we first employed quantitative real-time polymerase chain reaction (RT-qPCR) to study the gene expression levels related to lipid droplet formation and degradation. Figure 6A shows the effect of KIN on the lipid droplet formation-related genes in MDA-MB-231 cells, and the results indicated that the gene expression of DGAT1 was significantly inhibited. KIN exhibited the same effect on MDA-MB-468 cells (Figure 6B). Further evaluation revealed that KIN had no significant effect on the expression levels of genes involved in lipid droplet degradation, including HSL, ATGL, and MAGL (Figure 6C,D). The Western blotting assay also demonstrated that KIN significantly inhibited the protein expression level of DGAT1 in TNBC cells (Figure 6E,F). Compared to the control group, KIN could also inhibit the protein expression level of DGAT1 in vivo significantly (Figure 6G). The IHC experiment further confirmed the inhibitory effect of KIN on DGAT1 (Figure 6H). These results suggested that KIN inhibited lipid droplet formation in TNBC cells by suppressing DGAT1.

### 2.7. DGAT1 Mediates KIN-Induced Ferroptosis in TNBC Cells

In this study, we found that KIN significantly reduced the expression of DGAT1 both in vitro and in vivo. In order to examine whether DGAT1 plays a key role in modulating ferroptosis sensitivity, we conducted KIN overexpression experiments. DGAT1 overexpression significantly increased the protein expression level of DGAT1 in TNBC cells (Figure 7A,B). We found that DGAT1 overexpression significantly reduced the sensitivity of TNBC cells to KIN (Figure 7C,D). Figure 7E,F shows that DGAT1 overexpression significantly reduced the inhibitory effect of KIN on the GPX4 protein in TNBC cells. As in our previously presented data, KIN significantly reduced the level of GSH in TNBC cells, but this effect was mitigated by the overexpression of DGAT1 (Figure 7G,H). After DGAT1 was overexpressed in TNBC cells, the effect of KIN in increasing iron ion levels was reduced significantly (Figure 7I,J). Additionally, after the overexpression of DGAT1, the lipid ROS levels affected by KIN were partially restored (Figure 7K–N). The aforementioned evidence indicated that KIN promoted ferroptosis in TNBC cells by inhibiting DGAT1 expression.

## 3. Discussion

Previous studies have shown that KIN, a glycoside compound derived from *Anoectochilus roxburghii*, possesses a wide range of pharmacological activities [23,24]. In recent years, researchers have paid attention to the potential of KIN in treating diseases by regulating lipid metabolism. Lee et al. reported that KIN activated hormone-sensitive lipase and perilipin to facilitate lipolysis and to reduce fat accumulation, while also involving AMP-activated protein kinase activation in the process [18]. In addition, previous studies have confirmed that KIN can reduce ethanol-induced lipid accumulation in AML12 cells in a concentration-dependent manner [21]. However, whether KIN can treat TNBC by regulating lipid metabolism remains unclear. In this study, we found that KIN is an inhibitor of lipid droplet formation and exhibits effective activity against TNBC both in vitro and in vivo. KIN induced ferroptosis in TNBC cells by inhibiting lipid droplet formation, whereas the overexpression of the key lipid droplet formation protein DGAT1 resulted in resistance to KIN-induced ferroptosis in TNBC cells.

Firstly, we have established that KIN is an effective ferroptosis inducer in TNBC. Ferroptosis is associated with a variety of human diseases, such as cancer, cardiovascular disease, and inflammation [25]. The significant role of ferroptosis in tumor suppression makes it an attractive target for drug development. The process of ferroptosis is characterized by three basic features: iron accumulation, loss of the ability to repair lipid peroxides, and oxidation of membrane phospholipids containing PUFAs [26]. In this work, we confirmed the role of KIN in inducing ferroptosis in TNBC. Oxidative stress-induced ferroptosis contributes to the inhibition of tumor growth. Consistently, we found that KIN significantly reduced GSH levels while promoting an increase in lipid ROS levels. This confirms the role of KIN in regulating oxidative stress in TNBC. The potential mechanisms of ferroptosis are related to iron, amino acid, and lipid metabolism. PUFAs play a crucial role in promoting ferroptosis. Activated PUFAs bind to coenzyme A (CoA) to form long-chain acyl-CoA under the action of ACSL4. These changes can significantly induce ferroptosis [27]. In this study, we also found that KIN significantly increased the levels of PUFAs and ACSL4. This indicates that the mechanism by which KIN induces ferroptosis is closely related to lipid metabolism.

LD serves as the primary storage sites for neutral lipids within cells, characterized by their metabolically active and dynamic nature [28]. The dynamic changes in LD (formation, fusion, growth, and degradation) are closely associated with the onset of various diseases [29,30]. LD plays a crucial role in lipid and energy homeostasis. Recently, the regulatory role of LD in ferroptosis has also garnered attention. LD metabolism is involved in the regulation of ferroptosis through various pathways, such as buffering intracellular potentially toxic lipids, which plays a significant role in preventing lipid toxicity and oxidative stress. Additionally, LD can isolate intracellular PUFAs to regulate the occurrence of ferroptosis [31,32]. In the context of disease, lipid droplets serve a protective function in tumor cells by increasing PUFAs or inhibiting lipid droplet synthesis, leading to a decrease in cytoplasmic triglyceride levels, accompanied by an increase in ferroptosis levels [33]. In our study, we found that KIN has an excellent inhibitory effect on lipid droplet formation. DGAT1 is an essential enzyme in lipid droplet synthesis; the absence of DGAT1 in cells leads to a blockage in lipid droplet formation [34]. Subsequently, we used a lipid droplet inducer and conducted experiments with the overexpression of the lipid droplet formation protein DGAT1 to confirm that KIN induces ferroptosis by inhibiting lipid droplet formation. These data also confirm the tight connection between lipid droplets and ferroptosis. The regulation of PUFAs by lipid droplets is not only manifested in lipid droplet formation but also in degradation, which is a process that cannot be ignored. Therefore, we simultaneously tested genes related to both lipid droplet formation and degradation, and the results revealed that KIN only regulated lipid droplet formation. This also indicates the specificity of the mechanism of action of the monomeric compound.

As the in vivo experimental findings demonstrate, KIN can significantly inhibit tumor progression without causing liver or kidney toxicity, suggesting its potential as a therapeutic agent for TNBC patients. Since this study indicates that KIN has the properties of high efficacy and low toxicity, we hope that KIN can be developed as a potential therapeutic drug in future clinical trials. In addition, inhibiting the formation of LD to induce ferroptosis could potentially become a new and effective approach for cancer therapy. The molecular mechanisms underlying the interaction between ferroptosis and lipid droplets warrant further investigation.

## 4. Materials and Methods

### 4.1. Reagents

KIN was purchased from MCE (Cat No. HY-N2292, Princeton, NJ, USA). The MTT reagent and OA were acquired from Sigma-Aldrich (St. Louis, MO, USA). Kits for bicinchoninic acid (BCA) protein assays were supplied by the Beyotime Institute of Biotechnology in Shanghai, China. TC, TG, GSH, CRE, AST, BUN, and ALT determination kits were purchased from Nanjing Jiancheng Bioengineering Research Institute (Nanjing, China). Oil Red O was purchased from Solarbio Science & Technology (Beijing, China). The BODIPY493/503 dye (#D3922) used for immunofluorescence labeling was acquired from Thermo Fisher Scientific (Waltham, MA, USA), while 4′,6-Diamidino-2-phenylindole (DAPI) was sourced from Sigma-Aldrich (St. Louis, MO, USA). Antibodies against SLC7A11 (ab175186), GPX4 (ab125066), and DGAT1 (ab181180) were purchased from Abcam (Cambridge, UK). Antibodies against GAPDH (#5174) and ACSL4 (#38493) were purchased from Cell Signaling Technology (Danvers, MA, USA). C11-BODIPY and DFO were obtained from Selleck (Houston, TX, USA, AABF-H). The HE staining kits were purchased from Shanghai Biyuntian Biological Co., Ltd. (Shanghai, China).

### 4.2. In Vivo Tumor Xenograft Study

Healthy female BALB/c nude mice aged 4 – 6 weeks were obtained from Shanghai Nanfang Model Biotechnology Co., Ltd. (Shanghai, China) and maintained in a pathogen-free facility with a 12 h light/dark cycle, provided with sufficient food and water. In total, 2.5 × 10^6^ MDA-MB-231 cells, diluted in 100 μL PBS, were injected into the right flank of each mouse. When the tumors reach a size of 50–100 mm^3^, the mice were randomly divided into the blank control group (CON), the 15 mg/kg KIN group (KINL), the 30 mg/kg KIN group (KINH), and the 2 mg/kg doxorubicin positive control group (DOX), with 6 animals in each group. The mice’s weight and tumor volume were measured every other day. The drugs were administered intraperitoneally at the specified doses daily (the KIN suspension was formulated with 0.9% normal saline, while the CON was administered with 0.9% normal saline as the vehicle control), for a duration of 21 days.

### 4.3. HE Staining and IHC Analysis

Tumor xenografts, as well as liver and kidney specimens, were harvested, preserved in formalin for a night, and then embedded in paraffin wax. IHC staining was conducted on 5 μm sections with antibodies specific for DGAT1 (diluted 1:500) and Ki67 (diluted 1:200). For HE staining, the tissue sections were processed with this dye combination.

### 4.4. Cell Lines and Cell Culture

Cell lines, including MDA-MB-231, MDA-MB-468, and MCF-10A, were acquired from the Cell Bank at the Chinese Academy of Sciences in Shanghai, China, with the cells having undergone recent verification through STR profiling. MDA-MB-231 and MDA-MB-468 cells were cultured in DMEM medium supplemented with 10% FBS. MCF-10A cells were cultured in MCF-10A specialized medium with 10% FBS. Cell cultures were provided with a 1% penicillin/streptomycin solution and maintained at 37 °C under a 5% CO_2_ humidified environment.

To treat the cells, MDA-MB-231 and MDA-MB-468 cell lines were added with or without different concentrations (40 and 60 μM) of KIN for 48 h.

### 4.5. Cell Viability Assay

The MTT assay was employed to evaluate cell viability. In summary, the cells were plated at a rate of 1 × 10^4^ cells per well and treated with various compounds at specified concentrations for 48 h. Following incubation, 20 µL of MTT was added to each well, and after 4 h, DMSO was used to solubilize the formed formazan crystals. The absorbance was then recorded at 570 nm using a microplate reader (Bio-Tek, Santa Clara, CA, USA). The MTT assay was conducted three times, with each run consisting of 10 duplicates.

### 4.6. Colony Formation Assay

The cells were placed in 24-well plates and, once adhered, the culture medium was substituted with one that included various medications. After a two-week period, the medium was discarded, and the cells were subjected to two washes with phosphate-buffered saline (PBS), followed by fixation in 4% paraformaldehyde for a quarter of an hour. The cells were then stained in the dark with 0.1% crystal violet for 20 min, rinsed with distilled-deionized water, and allowed to air-dry at room temperature.

### 4.7. Oil Red O Staining

Intracellular lipid accumulation was quantified using Oil Red O staining. In summary, MDA-MB-231 and MDA-MB-468 cells, at a density of 2 × 10^5^ per well, were plated into 6-well dishes. Briefly, 2 × 10^5^ cells/well of MDA-MB-231 and MDA-MB-468 cells were seeded in 6-well plates. Upon reaching 60% confluence, the cells were added with different concentrations of KIN (0, 40, or 60 μM) for 48 h. The MDA-MB-231 and MDA-MB-468 cell lines were subjected to a double wash with PBS and then fixed in 4% formaldehyde for a duration of 10 min. Subsequently, the cells were stained with an Oil Red O solution consisting of 40% water and 60% isopropanol for 40 min, after which they were thoroughly rinsed with 60% isopropanol and imaged under a microscope. Following this, the red dye was eluted using 100% isopropanol, and the absorbance was read at 490 nm on a microplate reader. The data were normalized to the total protein content within the cells. The entire process was conducted three times, with two experimental replicates per run.

### 4.8. BODIPY 493/503 Fluorescence Staining

For BODIPY staining, the MDA-MB-231 and MDA-MB-468 cells were plated into laser confocal culture dishes at a rate of 1 × 10^4^ cells per well. When cell density reached about 60%, the cells were added with various concentrations of KIN (0, 40, or 60 μM) for a period of 48 h. The cells were subsequently rinsed twice with PBS to eliminate any remaining paraformaldehyde, followed by the addition of Hank’s balanced salt solution. Then, 3 mL of 1 μM BODIPY 493/503 was introduced to the cells and incubated at 37 °C for 30 min in the dark. The samples were subjected to three washes with PBS and promptly imaged using a laser confocal microscope. The overall cell fluorescence was quantified with ImageJ software version 1.54. This procedure was carried out three times, with two replicates for each trial.

### 4.9. Western Blotting

Western blotting was performed as described previously [35]. Cell lysates or frozen tumor specimens were disrupted using RIPA buffer on ice for half an hour. Following centrifugation at 16,000× *g* and 4 °C for a quarter of an hour, the supernatant was harvested, and the BCA protein assay kit (Beyotime, Shanghai, China) was utilized to quantify protein concentrations.

### 4.10. Biomedical Measurement

The levels of TC, TG, GSH, and total iron in the cells and the levels of ALT, AST, CRE, BUN, GSH, and total iron in the serum were determined using a SpectraMax Plus 384 Microplate Reader by Molecular Devices, located in Sunnyvale, CA, USA, following the manufacturer’s guidelines.

### 4.11. RNA-Sequencing (RNA-Seq)

In this study, MDA-MB-231 cells were treated with KIN or PBS (as a control) for 48 h and then the cell samples were, respectively, collected and analyzed by RNA-Seq implemented by Ringpu Bio-technology Co., Ltd. (Hangzhou, China).

### 4.12. RT-qPCR

According to the manufacturer’s instructions [36], RNA was extracted using Trizol reagent (TaKaRa, Shiga, Japan), and cDNA libraries were prepared with the Evo M-MLV RT Kit (AG11706, Accurate Biology, Guangzhou, China). RT-qPCR was carried out on a LightCycler 480II system (Roche, Basel, Switzerland) with the SYBR Green Premix Pro Taq HS qPCR Kit (AG11701, Accurate Biology). The primer sequences are provided in Appendix A. Relative gene expression was determined using the ΔΔCT method.

### 4.13. Measurement of Lipid ROS

A total of 1 × 10^5^ cells were plated into each well of a six-well plate. Following exposure to different concentrations of KIN for the specified duration, the cells were converted into a single-cell suspension through filtration and assessed using the Guava Easy Cytometer (Guava Technologies; Merck KGaA, Darmstadt, Germany).

### 4.14. Lentiviral-Mediated DGAT1 Overexpression

The plasmid containing the insert DGAT1 was obtained from OBIO Tech (Shanghai, China). Lentiviruses were produced by the co-transfection of the lentiviral vector with packaging plasmids into 293FT cells using Lipofectamine 2000 (Thermo Fisher Scientific), and transfection of the target cells was conducted as described previously [37].

### 4.15. PUFAs Quantification

PUFAs quantification was performed as described previously [38]. Briefly, 2 × 10^5^ cells/well of MDA-MB-231 and MDA-MB-468 cells were seeded in 6-well plates. Upon reaching 60% confluence, the cells were added with different concentrations of KIN (0, 40, or 60 μM) for 48 h. The cells collected were used to determine the fatty acid concentration.

### 4.16. Statistical Analysis

All data processing was conducted using GraphPad Prism 8.0, with the results displayed as mean ± standard deviation. For comparing two groups, Student’s t-tests were applied, and for multiple group comparisons, a one-way ANOVA test was used. Variability was assessed for each dataset. *: *p* < 0.05; **: *p* < 0.01; ***: *p* < 0.001; ^###^: *p* < 0.001; ns: *p* > 0.05.

## 5. Conclusions

In summary, KIN showed significant inhibitory effects on TNBC both in vivo and in vitro by triggering ferroptosis through the suppression of LD formation. We can confidently infer that KIN is a promising therapeutic agent for TNBC. Modulating the sensitivity to ferroptosis by intervening in the lipid droplet metabolism process as a treatment for diseases is an innovative therapeutic strategy.

## Figures and Tables

**Figure 1 ijms-26-02322-f001:**
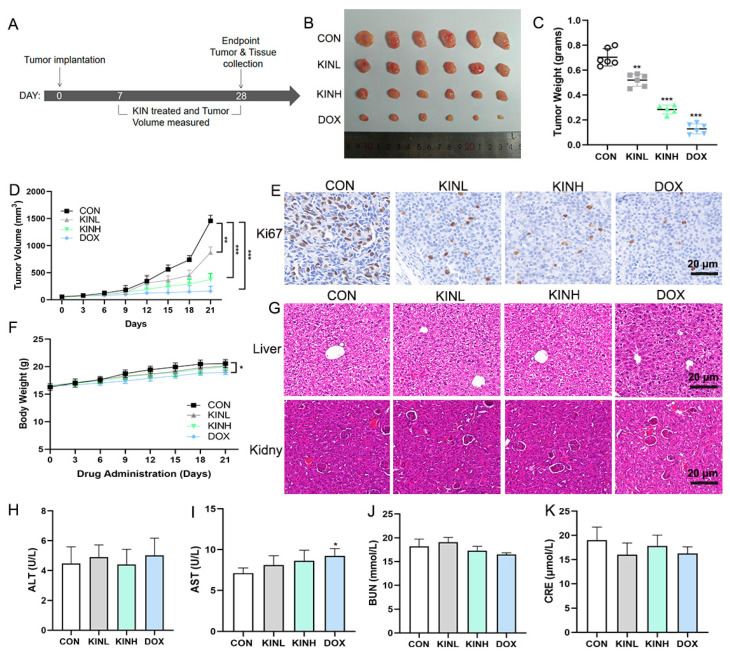
KIN inhibited tumor growth in mouse xenograft tumor models. (**A**) Timeline for tumor studies. (**B**) Representative images of tumors. (**C**,**D**) Tumor weight and volume were measured. (**E**) Representative IHC staining of Ki67 from sections of xenografted tumors; scale bar, 20 μm. (**F**) Body weight changed during treatment with drugs. (**G**) Representative Hematoxylin and eosin (HE) staining of liver (scale bar, 20 μm) and kidney (scale bar, 20 μm) sections from each group of mice. (**H**–**K**) Determination of ALT, AST, BUN, and CRE in serum of each group of mice. KIN, Kinsenoside; CON, control group; KINL, 15 mg/kg KIN group; KINH, 30 mg/kg KIN group; DOX, 2 mg/kg doxorubicin positive control group. *N* = 6 in each group. * *p* < 0.05, ** *p* < 0.01, *** *p* < 0.001, * vs. CON group.

**Figure 2 ijms-26-02322-f002:**
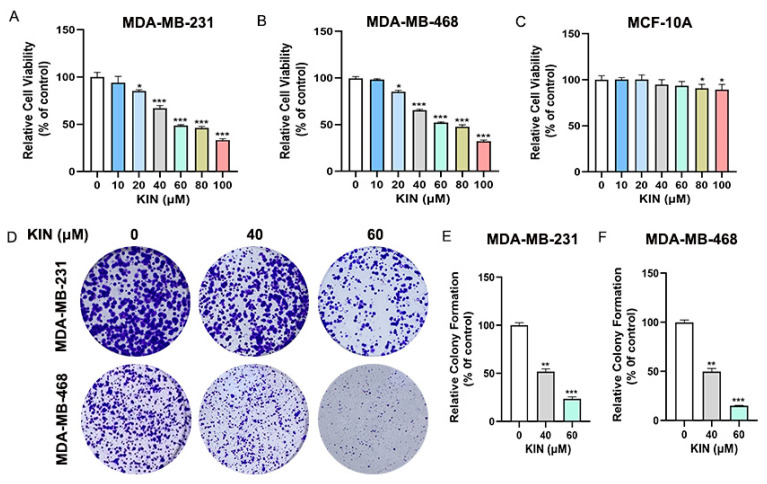
KIN inhibits growth of TNBC cells in vitro. (**A**–**C**) MDA-MB-231, MDA-MB-468, and MCF-10A cells were treated with KIN for 48 h, and cell viability was measured by MTT assay. (**D**) MDA-MB-231 and MDA-MB-468 cells were treated with KIN for 2 weeks, and colony formation of cells was detected by crystal violet staining. (**E**,**F**) Quantification of colony numbers from ImageJ. KIN, Kinsenoside. Data are representative of three independent experiments. * *p* < 0.05, ** *p* < 0.01, *** *p* < 0.001.

**Figure 3 ijms-26-02322-f003:**
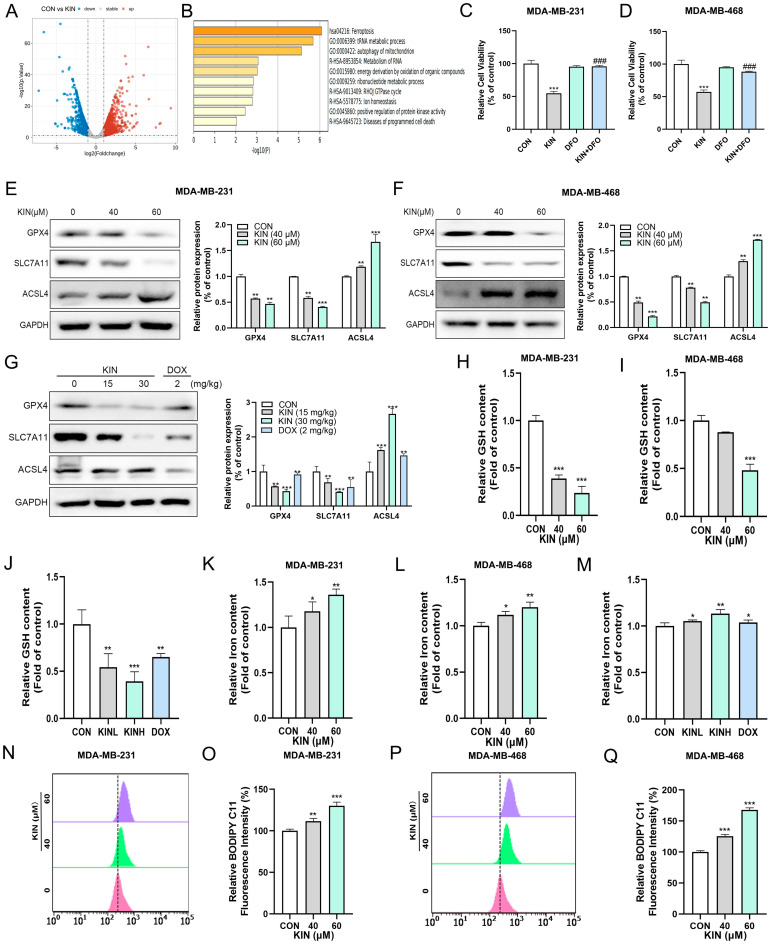
KIN-induced ferroptosis in TNBC cells. (**A**) Volcano plot of downregulated and upregulated genes in control and KIN treatment MDA-MB-231 cells. (**B**) KEGG enrichment analysis based on differentially expressed genes. (**C**) Ferroptosis inhibitors used in MDA-MB-231. (**D**) Ferroptosis inhibitors used in MDA-MB-468. (**E**) Ferroptosis pathway protein levels in MDA-MB-231 cells after KIN treatment were detected. (**F**) Ferroptosis pathway protein levels in MDA-MB-468 cells after KIN treatment were detected. (**G**) Ferroptosis pathway protein levels in animal experiments after KIN treatment were detected. (**H**–**J**) GSH levels in vitro and in vivo were detected by GSH kit. (**K**–**M**) Total iron levels in vitro and in vivo were detected by iron detection kit. (**N**–**Q**) Lipid peroxidation levels were evaluated by C11-BODIPY fluorescent probe. Data are representative of three independent experiments. KIN, Kinsenoside; DFO, deferoxamine; CON, control group; KINL, 15 mg/kg KIN group; KINH, 30 mg/kg KIN group; DOX, 2 mg/kg doxorubicin positive control group. * *p* < 0.05, ** *p* < 0.01, *** *p* < 0.001, ^###^
*p* < 0.001, * vs. CON group, ^#^ vs. KIN group.

**Figure 4 ijms-26-02322-f004:**
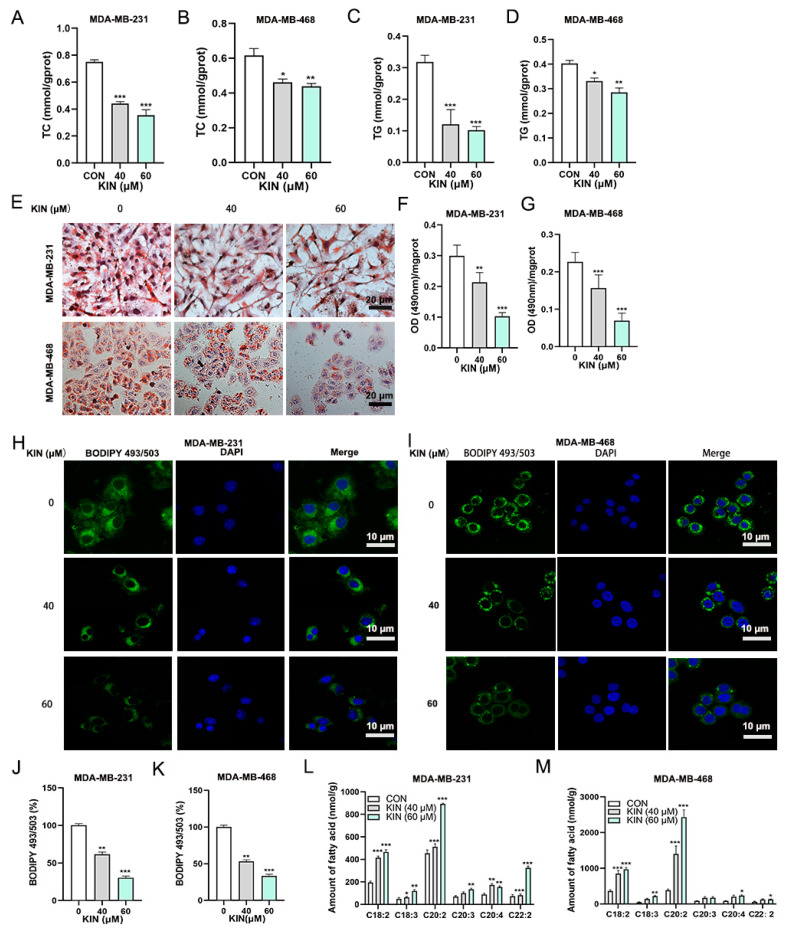
KIN inhibits lipid droplets formation in TNBC cells. MDA-MB-231 and MDA-MB-468 cells were treated with KIN for 48 h, and intracellular levels of TC (**A**,**B**), and TG (**C**,**D**) were evaluated using commercial kits. (**E**) Oil Red O staining of lipid droplet accumulation in TNBC cells, scale bar, 20 μm. (**F**,**G**) Quantitative analysis of Oil Red O staining in TNBC cells. BODIPY 493/503 fluorescence (green) staining of lipid droplet formation in MDA-MB-231 (**H**) and in MDA-MB-468 (**I**) cells; scale bar, 10 μm. Quantification of BODIPY 493/503 fluorescent intensity in MDA-MB-231 (**J**) and MDA-MB-468 (**K**) cells. Effect of KIN on levels of PUFAs in MDA-MB-231 (**L**) and MDA-MB-468 (**M**) cells was detected by mass spectrometry. Data are representative of three independent experiments. KIN, Kinsenoside; CON, control group. * *p* < 0.05, ** *p* < 0.01, *** *p* < 0.001.

**Figure 5 ijms-26-02322-f005:**
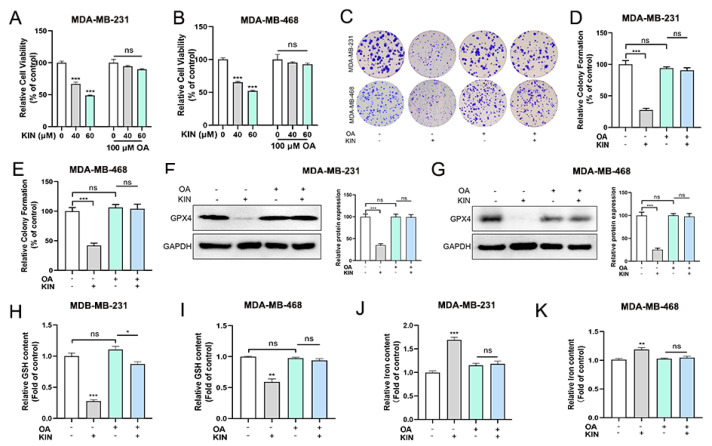
KIN induces ferroptosis in TNBC cells by inhibiting lipid droplet formation. (**A**,**B**) MDA-MB-231 and MDA-MB-468 cells were treated with KIN and OA for 48 h, and cell viability was measured by MTT assay. (**C**) MDA-MB-231 and MDA-MB-468 cells were treated with KIN and OA for 2 weeks, and colony formation of cells was detected by crystal violet staining. (**D**,**E**) Quantification of colony numbers from Image J. GPX4 protein levels in MDA-MB-231 (**F**) and MDA-MB-468 (**G**) cells after KIN and OA treatment were detected. (**H**,**I**) GSH levels in vitro were detected by GSH kit. (**J**,**K**) Total iron levels in TNBC cells after KIN and OA treatment were detected by iron detection kit. Data are representative of three independent experiments. KIN, Kinsenoside; OA, oleic acid. * *p* < 0.05, ** *p* < 0.01, *** *p* < 0.001, ns *p* > 0.05.

**Figure 6 ijms-26-02322-f006:**
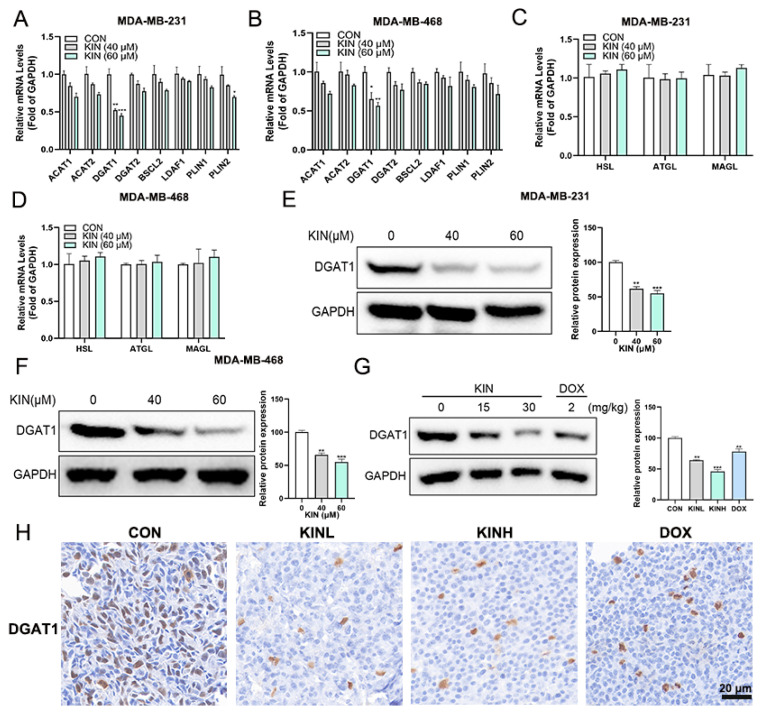
KIN inhibits lipid droplet formation by inhibiting DGAT1. (**A**,**B**) Gene expression levels related to lipid droplet formation were measured by RT-qPCR. (**C**,**D**) Gene expression levels related to lipid droplet degradation were measured by RT-qPCR. DGAT1 protein levels in MDA-MB-231 (**E**) and MDA-MB-468 (**F**) cells after KIN treatment were detected. (**G**) DGAT1 protein levels in animal experiments after KIN treatment were detected. (**H**) Representative IHC staining of DGAT1 from sections of xenografted tumors; scale bar, 20 μm. Data are representative of three independent experiments. KIN, Kinsenoside; CON, control group; KINL, 15 mg/kg KIN group; KINH, 30 mg/kg KIN group; DOX, 2 mg/kg doxorubicin positive control group. * *p* < 0.05, ** *p* < 0.01, *** *p* < 0.001.

**Figure 7 ijms-26-02322-f007:**
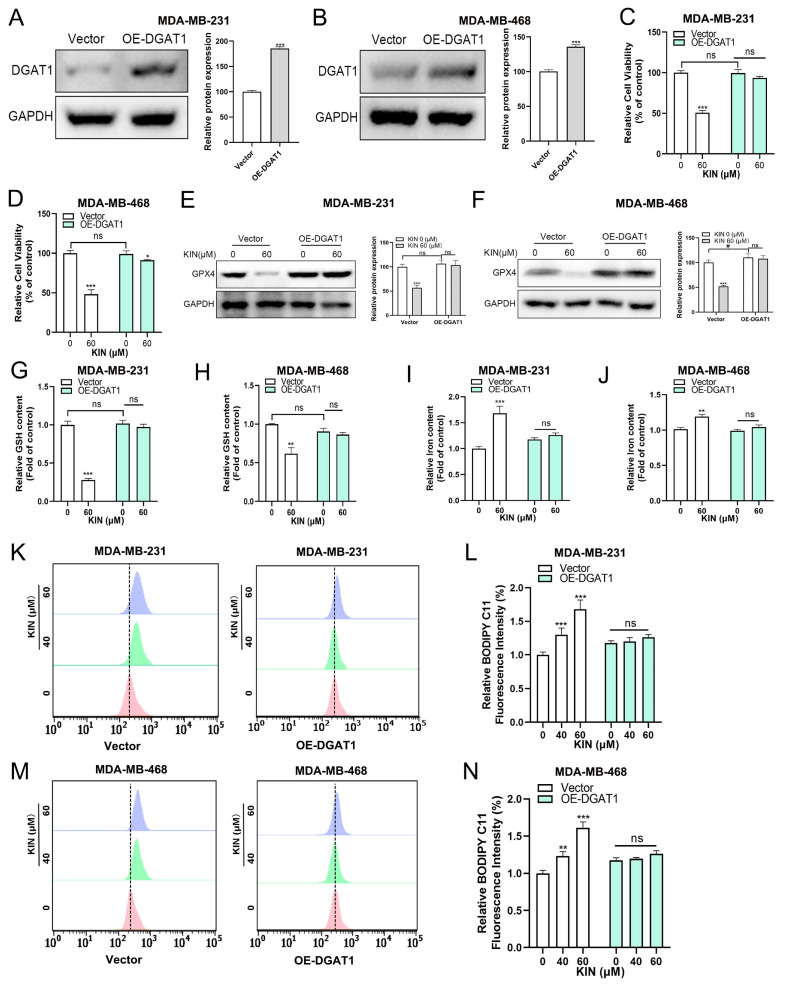
DGAT1 mediates KIN-induced ferroptosis in TNBC cells. (**A**,**B**) Protein levels of DGAT1 were detected in TNBC cells transfected with vector or OE-DGAT1. (**C**,**D**) TNBC cells transfected with vector or OE-DGAT1 were treated with KIN for 48 h and cell viability was measured by MTT assay. GPX4 protein levels in MDA-MB-231 (**E**) and MDA-MB-468 (**F**) cells were detected. (**G**,**H**) GSH levels in TNBC cells were detected by GSH kit. (**I**,**J**) Total iron levels in TNBC cells were detected by iron detection kit. (**K**–**N**) Lipid peroxidation levels were evaluated by C11-BODIPY fluorescent probe. Data are representative of three independent experiments. KIN, Kinsenoside. * *p* < 0.05, ** *p* < 0.01, *** *p* < 0.001, ^#^
*p* < 0.05, ns *p* > 0.05.

## Data Availability

The data that support the findings of this study are available from the corresponding author upon reasonable request.

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
