# Peer review of "Kinsenoside Suppresses DGAT1-Mediated Lipid Droplet Formation to Trigger Ferroptosis in Triple-Negative Breast Cancer"

_ijms, 2025, doi:10.3390/ijms26052322_

Round 1

Reviewer 1 Report

Comments and Suggestions for Authors

Authors in their work investigate the therapeutic effects of Kinsenoside (KIN) on triple-negative breast cancer (TNBC). KIN inhibits tumor growth in vivo and significantly reduces the viability and proliferation of TNBC cells in vitro. KIN also decreases lipid droplet (LD) formation and lipid content in these cells. Through transcriptomics and inhibitor-rescue experiments, the researchers reveal that KIN induces ferroptosis  in TNBC cells. This process is marked by increased lipid peroxidation, iron accumulation, and glutathione (GSH) depletion. Additionally, overexpression of DGAT1, a protein involved in lipid droplet formation, mitigates KIN’s effects on cell viability and ferroptosis induction. KIN's impact on tumor growth occurs without significant toxicity to the liver or kidneys. These findings suggest that KIN may offer a promising therapeutic approach for TNBC by targeting ferroptosis through the suppression of DGAT1-mediated lipid droplet formation. Study is very well planned and presented. It uses both in-vitro and in-vivo models providing clear explanation of the cytostatic effect of KIN at the molecular level. I have just one suggestion, it is worth to provide in the descriptions of the figures all the date which allows to understand the figure without looking for references in the text. Therefore please include in the figures descriptions expansion of abbreviations: KINL, KINH, DOX, etc.

Author Response

Comments 1: Authors in their work investigate the therapeutic effects of Kinsenoside (KIN) on triple-negative breast cancer (TNBC). KIN inhibits tumor growth in vivo and significantly reduces the viability and proliferation of TNBC cells in vitro. KIN also decreases lipid droplet (LD) formation and lipid content in these cells. Through transcriptomics and inhibitor-rescue experiments, the researchers reveal that KIN induces ferroptosis  in TNBC cells. This process is marked by increased lipid peroxidation, iron accumulation, and glutathione (GSH) depletion. Additionally, overexpression of DGAT1, a protein involved in lipid droplet formation, mitigates KIN’s effects on cell viability and ferroptosis induction. KIN's impact on tumor growth occurs without significant toxicity to the liver or kidneys. These findings suggest that KIN may offer a promising therapeutic approach for TNBC by targeting ferroptosis through the suppression of DGAT1-mediated lipid droplet formation. Study is very well planned and presented. It uses both in-vitro and in-vivo models providing clear explanation of the cytostatic effect of KIN at the molecular level. I have just one suggestion, it is worth to provide in the descriptions of the figures all the date which allows to understand the figure without looking for references in the text. Therefore please include in the figures descriptions expansion of abbreviations: KINL, KINH, DOX, etc.

Response 1: We sincerely appreciate the valuable comments. We have taken your advice into account and have made the necessary revisions to the manuscript. Specifically, we have expanded the abbreviations in the figure descriptions as you recommended. Now, the descriptions include the full expansions of abbreviations such as KINL, KINH, DOX, and others. This should enable readers to understand the figures without needing to refer back to the text for clarification. We believe these changes have enhanced the clarity and completeness of our figures, making the data more accessible to the readers. We appreciate your input and are grateful for the opportunity to improve our manuscript (page 3, lines 97-99; page 4, line 116; page 5, lines 151-152; page 7, lines 182; page 8, lines 207; page 9, lines 230-231; page 10, line 255).

Reviewer 2 Report

Comments and Suggestions for Authors

The therapeutic potential of Kinsenoside (KIN) in triple-negative breast cancer (TNBC) and its molecular mechanisms are investigated in the manuscript. The authors confirmed in vivo and in vitro experiments that KIN inhibits TNBC cell growth by inhibiting DGAT1-mediated lipid droplet formation and inducing ferroptosis. The results showed that KIN could significantly inhibit tumor growth without significant toxicity to liver and kidney. In addition, KIN promotes lipid peroxidation and iron accumulation by increasing levels of polyunsaturated fatty acids (PUFAs), ultimately leading to iron death in TNBC cells. The study also further verified the mechanism of KIN through transcriptomics and inhibitor rescue experiments. The manuscript logic is reasonable and clear, but it needs to be modified due to the following problems.

 Main comments:

  1. The background of TNBC and Iron death is fully introduced in the introduction, but it can be further simplified and focused to avoid too much background information interfering with readers' attention to the core content. The description of the result part is more detailed, but some paragraphs can be further simplified to avoid repeated description. For example, some of the descriptions in Figures 3 and 4 can be combined to reduce redundancy.
  2. It is suggested that the potential application of KIN in other types of cancer be further explored in the discussion section, in particular whether it could play a role in other cancers through a similar mechanism (inhibiting lipid droplet formation to induce iron death).
  3. For the toxicity test of KIN, although the authors have carried out pathological examination of liver and kidney, it is recommended to add more long-term toxicity test data to further verify the safety of KIN. 4. The layout in the figure is rather chaotic, so it is suggested that the figure of different letters can be properly combined into one, and the statistics of F as E in Figure 3 should be combined with E to represent it as a letter, and the picture layout is chaotic, and there are such problems in Figure 4, Figure 5, Figure 6 and Figure 7. It is suggested to reorganize the figure.

 Minor comments:

  1. Improve the resolution of some images, especially Western blot strips. 2. Some sentences have grammatical errors or are not concise enough. It is recommended to polish the text, especially the results and discussion sections.
  2. In Figure 3-7, the font size needs to be adjusted.

Author Response

Comments 1: The background of TNBC and Iron death is fully introduced in the introduction, but it can be further simplified and focused to avoid too much background information interfering with readers' attention to the core content. The description of the result part is more detailed, but some paragraphs can be further simplified to avoid repeated description. For example, some of the descriptions in Figures 3 and 4 can be combined to reduce redundancy.

Response 1: Thank you for this helpful suggestion. We have simplified the background of TNBC and ferroptosis in the introduction, focusing more on the core content and reducing background details to enhance clarity (page 1, lines 38-40). In addition, we have carefully checked the descriptions in the results section, combining some of the descriptions for Figures 3 and 4 to reduce redundancy (page 4, lines 134-136; page 6, lines 156-158; lines 162-163; lines 164-166).

Comments 2: It is suggested that the potential application of KIN in other types of cancer be further explored in the discussion section, in particular whether it could play a role in other cancers through a similar mechanism (inhibiting lipid droplet formation to induce iron death).

Response 2: We sincerely appreciate the valuable comments. Kinsenoside (KIN) is a major active compound in Anoectochilus roxburghii. Currently, only a limited number of studies have reported on the anti-tumor activity of Anoectochilus roxburghii extracts. Researchers have discovered that Anoectochilus roxburghii possesses remarkable anti-tumor effects, including against breast cancer and colon cancer. These have been described in detail in our Introduction section (page 2, lines 62-66). To further enhance the discussion, we have incorporated recent evidence demonstrating KIN's therapeutic potential in disease management through the regulation of lipid metabolism. Moreover, we propose that the inhibition of lipid droplet formation to induce ferroptosis could represent a novel therapeutic strategy for tumor treatment. The above information has been added to the Discussion section of the manuscript (page 10, lines 259-266; page 12, lines 314-316).

Comments 3: For the toxicity test of KIN, although the authors have carried out pathological examination of liver and kidney, it is recommended to add more long-term toxicity test data to further verify the safety of KIN.

Response 3: Thanks for your kind and professional suggestion. In response to the reviewer’s comment regarding the toxicity test of KIN, we acknowledge the importance of comprehensive long-term toxicity data in establishing the safety profile of KIN. While we have included pathological examinations of the liver and kidney in our current study, we fully agree with the suggestion to incorporate additional long-term toxicity test results. In this study, we conducted in vivo experiments using 15 mg/kg and 30 mg/kg of KIN, and our experiments confirmed that no toxicity was observed. Currently, there are studies that have used 100 mg/kg of KIN for in vivo research, and no significant toxicity has been found (Phytomedicine 2019, 55, 255-263). These data to some extent confirm that KIN is a safe and effective compound. However, we fully agree with the reviewer's opinion. We plan to conduct these extended studies in the near future and will include the findings in our subsequent publications. We believe that this will provide a more robust assessment of KIN’s safety, addressing the concerns raised by the reviewer. Thank you for your valuable input, which will undoubtedly enhance the quality of our research. 

Comments 4: The layout in the figure is rather chaotic, so it is suggested that the figure of different letters can be properly combined into one, and the statistics of F as E in Figure 3 should be combined with E to represent it as a letter, and the picture layout is chaotic, and there are such problems in Figure 4, Figure 5, Figure 6 and Figure 7. It is suggested to reorganize the figure.

Response 4: Thank you for your valuable feedback on our manuscript. We appreciate your insightful comments regarding the layout and representation of the figures. We have taken your suggestions seriously and have made significant revisions to address the issues you raised. Specifically, we have reorganized all the figures to ensure a more coherent and organized layout. Additionally, we have combined the figures of different letters where appropriate into a single figure, as you suggested. Regarding the issue you pointed out in Figure 3, we have corrected this mistake. We have now combined the data for ‘F’ and ‘E’ and represent them collectively under the letter ‘E’, as per your recommendation. Furthermore, we have examined Figures 4, 5, 6, and 7 and have applied the same principles of reorganization and combination where applicable. We believe that these revisions have significantly improved the clarity and accuracy of our figures, making them more comprehensible to the readers. We trust that these changes meet your expectations and address the concerns you had regarding the figure organization. Once again, we thank you for your constructive feedback, which has been instrumental in enhancing the quality of our manuscript.

Comments 5: Improve the resolution of some images, especially Western blot strips. 

Response 5: Thanks for your kind and professional suggestion. We have taken your comments seriously and have made the necessary improvements. Specifically, we have enhanced the resolution of some images, particularly the Western blot strips, to ensure better clarity and readability. We believe these adjustments address the concerns you raised and improve the overall quality of the manuscript.

Comments 6: Some sentences have grammatical errors or are not concise enough. It is recommended to polish the text, especially the results and discussion sections.

Response 6: Thank you for your careful checks. We have meticulously reviewed the entire manuscript and corrected any grammatical errors. Additionally, we have revised the Results and Discussion sections to enhance clarity and conciseness. We believe these changes have significantly improved the quality of our manuscript.

Comments 7: In Figure 3-7, the font size needs to be adjusted.

Response 7: We sincerely appreciate your valuable comments. We have addressed the issue regarding the font size in Figures 3-7. The font sizes have been adjusted to improve readability and consistency throughout these figures. We believe this enhancement will make the data presentation clearer and more accessible to the readers.

Round 2

Reviewer 2 Report

Comments and Suggestions for Authors

The authors have addressed the reviewer's concerns and the current version is acceptable.